# The Real-Life Impact of Primary Tumor Resection of Synchronous Metastatic Colorectal Cancer—From a Clinical Oncologic Point of View

**DOI:** 10.3390/cancers16081460

**Published:** 2024-04-11

**Authors:** Balázs Pécsi, László Csaba Mangel

**Affiliations:** Institute of Oncotherapy, Clinical Center and Medical School, University of Pécs, 7624 Pécs, Hungary

**Keywords:** metastatic colorectal carcinoma, primary tumor resection, overall survival

## Abstract

**Simple Summary:**

Colorectal cancer is one of the most frequent malignant diseases in the world. The question of whether to remove the primary tumor in asymptomatic synchronous metastatic colorectal cancer or keep the tumor intact is still controversial. The operation before the necessary systemic chemotherapy has many pros and cons. For those patients who are candidates for metastasectomy, the removal of the primary tumor is inevitable. We conducted a retrospective data analysis of 449 patients treated in six years. The study showed clear evidence that removing the primary tumor results in fewer primary-tumor-related complications (bleeding, obstruction) and significantly improves overall survival. The delay of systemic therapy due to the operation did not show any undesired consequences. We strongly think that for those who are suitable for any kind of manipulation of the primary tumor (removal or irradiation of the rectal cancer), this is undoubtedly advantageous.

**Abstract:**

Aim: The complex medical care of synchronous metastatic colorectal (smCRC) patients requires prudent multidisciplinary planning and treatments due to various challenges caused by the primary tumor and its metastases. The role of primary tumor resection (PTR) is currently uncertain; strong arguments exist for and against it. We aimed to define its effect and find its best place in our therapeutic methodology. Method: We performed retrospective data analysis to investigate the clinical course of 449 smCRC patients, considering treatment modalities and the location of the primary tumor and comparing the clinical results of the patients with or without PTR between 1 January 2013 and 31 December 2018 at the Institute of Oncotherapy of the University of Pécs. Results: A total of 63.5% of the 449 smCRC patients had PTR. Comparing their data to those whose primary tumor remained intact (IPT), we observed significant differences in median progression-free survival with first-line chemotherapy (mPFS1) (301 vs. 259 days; *p* < 0.0001; 1 y PFS 39.2% vs. 26.6%; OR 0.56 (95% CI 0.36–0.87)) and median overall survival (mOS) (760 vs. 495 days; *p* < 0.0001; 2 y OS 52.4 vs. 26.9%; OR 0.33 (95% CI 0.33–0.53)), respectively. However, in the PTR group, the average ECOG performance status was significantly better (0.98 vs. 1.1; *p* = 0.0456), and the use of molecularly targeted agents (MTA) (45.3 vs. 28.7%; *p* = 0.0005) and rate of metastasis ablation (MA) (21.8 vs. 1.2%; *p* < 0.0001) were also higher, which might explain the difference partially. Excluding the patients receiving MTA and MA from the comparison, the effect of PTR remained evident, as the mOS differences in the reduced PTR subgroup compared to the reduced IPT subgroup were still strongly significant (675 vs. 459 days; *p* = 0.0009; 2 y OS 45.9 vs. 24.1%; OR 0.37 (95% CI 0.18–0.79). Further subgroup analysis revealed that the site of the primary tumor also had a major impact on the outcome considering only the IPT patients; shorter mOS was observed in the extrapelvic IPT subgroup in contrast with the intrapelvic IPT group (422 vs. 584 days; *p* = 0.0026; 2 y OS 18.2 vs. 35.9%; OR 0.39 (95% CI 0.18–0.89)). Finally, as a remarkable finding, it should be emphasized that there were no differences in OS between the smCRC PTR subgroup and metachronous mCRC patients (mOS 760 vs. 710 days, *p* = 0.7504, 2 y OS OR 0.85 (95% CI 0.58–1.26)). Conclusions: The role of PTR in smCRC is still not professionally justified. Our survey found that most patients had benefited from PTR. Nevertheless, further prospective trials are needed to clarify the optimal treatment sequence of smCRC patients and understand this cancer disease’s inherent biology.

## 1. Introduction

Colorectal cancer (CRC) is the third most common cancer and the second leading cause of cancer-related deaths worldwide. The incidence of CRC is increasing every year, especially in people under 50 years of age. Due to innovative and more efficient healthcare methods and different screening projects, there has been a slight decrease in mortality, but still, half of the patients die of cancer [1]. 

Almost 30% of CRCs are diagnosed after the cancer has spread to distant tissues or organs. Among the primary nonmetastatic cases, 50–70% will develop a metastatic disease (mCRC) during the disease course. Patients with mCRC face poor prognosis in general, with a relative 5-year survival rate of 14–17% [2]. Despite a growing proportion of patients amenable to curative-intent resection or ablation, the aim of treatment in most patients with mCRC is palliation to optimize quality of life (QoL) and overall survival (OS). While surgery is the cornerstone treatment for early-stage cancer, chemotherapy is generally the first treatment option for metastatic disease. While the role of the resection of the primary tumor (PTR) is widely accepted in some emergencies or in potentially curable, e.g., oligometastatic patients, the PTR for non- or mildly symptomatic patients with no curative intent is still under debate. This issue has several pros and cons; no prospective randomized clinical trial with remarkable results could clarify this question [3]. 

Based on our large database, previously utilized for different real-world data cohort analyses [4,5], we tried to evaluate the real-world role of PTR in the treatment of synchronous metastatic colorectal cancer (smCRC). The study aimed to find the clinical situations where PTR may find its best place or where PTR should be avoided. We focused our research on the role of PTR in non- or mildly symptomatic patients who are not candidates for curative therapies, namely, not candidates for metastasectomy. Nevertheless, numerous advantages and disadvantages may be collected on both sides of the scale, and each patient probably requires individual decisions. Still, we hope our observations might assess basic information to help make optimal clinical decisions.

## 2. Method

We surveyed all data of the metastatic colorectal cancer patients treated at the Institute of Oncotherapy of the University of Pécs Clinical Centre between 1 January 2013 and 31 December 2018. We analyzed a lot of different patient-related, disease-specific and outcome parameters in every patient (e.g., patient-related data: age at mCRC diagnosis, sex, performance status; tumor-related data: localization; metastasis-related data: temporality (synchronous or metachronous), focality and localization of metastases; treatment-related data: primary tumor resection, type of chemotherapy; palliative radiotherapy of intrapelvic tumors; clinical outcome parameters: progression-free survival (PFS) and overall survival (OS), the reason of discontinuation of chemotherapy) [4,5].

Based on these data, the current work compares the clinical outcome of smCRC patients with (PTR) or without primary tumor resection (intact primary tumor—IPT). In some comparisons, the data of metachronous metastatic colorectal cancer patients (mmCRC) were used as a control arm, assuming its similarity to the PTR group as both groups were without primary tumors. Further subgroups were created on a basis of the localization of the primary tumor, such as intrapelvic and extrapelvic tumors. Extrapelvic primary tumors were divided into two embryologically different colon sides, such as right-sided (RCC) and left-sided (LCC) primary tumors. Further comparisons to reveal the actual value of PTR, reduced PTR, and IPT subgroups were made by excluding those factors that were significantly different in addition to the PTR, such as the use of molecularly targeted agents (MTA), metastasis ablation (MA), and patients’ performance status > ECOG 1.

The study’s primary objectives were the mPFS, mOS, rate of 1 y PFS, and rate of 2 y OS, depending on the different therapeutic approaches and tumor/metastasis status.

Descriptive statistics were used to characterize the patient cohorts. Differences in categorical parameters were analyzed using a two-sample t-test. The level of significance of *p* ≤ 0.05 was used. Progression-free and overall survival were estimated using the Kaplan–Meier method. The odds ratio was calculated within 95% confidence intervals.

It should be noted that at the time of data analysis (17 March 2022), 40 patients (7.8%) were still alive.

## 3. Results

Between 1 January 2013 and 31 December 2018, 664 patients received first-line treatment (chemotherapy +/− MTA) for mCRC at the Institute of Oncotherapy of Clinical Centre of the University of Pécs. A total of 67.6% (449 patients) had smCRC. Among them, in 63.5% (285 patients), PTR was performed, and 36.5% (164 patients) belonged to the IPT group. The general information about our patients and the data on their metastatic tumors are collected in Table 1.

The smCRC group was divided into two subgroups, the PTR and IPT groups; the third group, the mmCRC group, was used as a control arm in some comparisons. Considering the PTR and IPT subgroups, there were no significant differences in sex and age distribution, but the ECOG performance status (PS) reached a significant level (*p* = 0.0456) in favor of the PTR subgroup. Based on the localization and the findings, the mCRC patients were ultimately divided into two subgroups with intrapelvic primary tumors (rectum and lower sigmoid colon—rectosigmoid colon cancer, RSC) and with extrapelvic primary tumors (cecum, ascending colon, transverse colon—right-sided colon cancer, RCC, and descending colon and oral part of the sigmoid colon—left-sided colon cancer, LCC). This classification can also be theoretically explained by the treatment modality differences (intrapelvic tumors can be irradiated, and extrapelvic tumors are more accessible for resection). The extrapelvic primary tumor localization rate in the PTR subgroup was significantly higher (67.4% vs. 47.6%, *p* < 0.0001). Finally, and not surprisingly, the number of more than one involved metastatic organ was lower in the PTR subgroup (24.2 vs. 37.2%, *p* = 0.0034) (see Table 1).

All patients received systemic fluoropyrimidine-based (FP) (5-fluoro-uracil continuous infusion or oral capecitabine) treatment, and 11.6 and 13.4% received only FP monotherapy. The majority of patients received doublet therapy (FP + irinotecan or oxaliplatin) (FOLFIRI/CAPIRI or FOLFOX/CAPOX scheme) with or without MTA, either with vascular endothelial growth factor inhibitor (VEGFi) (bevacizumab) or epidermal growth factor receptor inhibitors (EGFRi) (cetuximab or panitumumab) (properly only in K-Ras wild type patients). Though there were no differences in using doublet therapy between the subgroups (88.4 vs. 86.6%; *p* = 0.5688), the application of MTA showed a significant difference (45.3% vs. 28.7%, *p* = 0.0005) (see Table 2).

The rate of ablation of metastases (MA) by any method (surgery 93.5%, radiofrequency or radiotherapy 3.2%, each) among the two main subgroups showed remarkable observable differences in favor of the PTR subgroup (21.8% vs. 1.2%, *p* < 0.0001) (see Table 2).

Considering the outcomes of the first-line chemotherapy ± MTA, a robust difference was proved in PFS1 and OS between the two main subgroups (*p* < 0.0001, both). However, it should be noted that the OS result in the mmCRC group was similar to the PTR subgroup (*p* = 0.7504, 2 y OS OR 0.85 (95% CI 0.58–1.26)) and showed the same degree of difference as the IPT subgroup (*p* < 0.0001, 2 y OS OR 0.39 (95% CI 0.24–0.64)) (see Table 3). The Kaplan–Meier curve for comparing OS in PTR and IPT subgroups is shown in Figure 1.

However, we found a significant difference between the two subgroups for the rate of mono- and multiorgan metastases (*p* = 0.0034); still, the proportion of different organ-specific metastases did not differ considerably: liver metastases (75.4% vs. 81.7%, *p* = 0.1699), lung metastases (20.7% vs. 26.2%, *p* = 0.1799) and peritoneal metastases (16.8% vs. 22.6%, *p* = 0.1370). In the control mmCRC group, this rate was 44.7%, 33.5%, and 13.5%, respectively, showing a more significant difference in the percentage of involved organs (see Table 4).

We also tried to find an outcome difference between extrapelvic primary tumors (RCC and LCC) based on the well-known unanalogous biology of the two embryologically different colon sides. We did not observe a significant survival difference in the PTR or IPT subgroup. In the PTR subgroup, the mOS values in the RCC and LCC primary tumors were 552 and 818 days (*p* = 0.3487, 2 y OS OR 0.42 (95% CI 0.21–0.83); and in the IPT subgroup, these values were 376 and 436 days (*p* = 0.8611, 2 y OS OR 0.64 (95% CI 0.81–2.32). Though there was a clear decreased survival trend by the well-known worse prognostic RCC group, it was far from the significant level in our database (see Table 5A). 

Meanwhile, when dividing the primary localization into extra- and intrapelvic subgroups, a significant OS disadvantage was found exclusively in the extrapelvic IPT subgroup compared to intrapelvic IPT (*p* = 0.0070) and extrapelvic PTR subgroups (*p* < 0.0001) (see Table 5B).

The comparison of each primary tumor localization in the PTR and IPT subgroups brought significant differences both in mPFS1 and mOS (RCC *p* = 0.0305 and *p* = 0.0069; LCC *p* < 0.0001 and *p* < 0.0001 and RSC *p* = 0.0049 and 0.0083). 

These data confirm the assumption of the positive prognostic impact of PTR in the survival of smCRC cases in our patient cohorts.

Conclusively, all major differences between the PTR and IPT subgroups of smCRC patients are collected in Table 6.

For the conclusion and to assess the real value of PTR, from our database, we excluded from both PTR and IPT subgroups all patients who received MTA and MA and whose PS was worse than ECOG 1. The results of these reduced subgroups are shown in Figure 2. It is clear that the survival curve of the PTR subgroup showed almost the same results as the control mmCRC group, but both groups differed significantly from the reduced IPT subgroup. 

PTR was carried out in emergency in 23.2% (urgent PTR patients), and elective surgical intervention was performed in 76.8% (elective PTR patients). Between these two subgroups, there were no significant differences in age, PS, and gender distribution. Significant or near-significant differences were found in the localization of the primary tumor (extrapelvic localization 86.4 vs. 61.6%, *p* = 0.0002) and the rate of applied MA (13.6 vs. 24.2%, *p* = 0.0686). There were no significant differences in mPFS1 (259 vs. 310 days, *p* = 0.3153), but mOS showed a significant difference (819 vs. 534 days, *p* = 0.0092; 2 y OS 58.0 vs. 33.9%; OR 0.37 (95% CI 0.19–0.71)). 

Excluding the MA-applied patients from these subgroups, there were no more significant differences in outcome between these subgroups (mPFS1 301 vs. 257 days; *p* = 0.7794; 1 y PFS OR 0.55 (95% CI 0.27–1.12) and mOS 707 vs. 483 days; *p* = 0.0886; 2 y OS OR 0.47 (95% CI 0.23–0.95)). 

Comparing the effect of MA within the urgent and elective PTR patients, a significant difference in mOS was found only in the elective PTR subgroup (*p* = 0.0009), not in the urgent PTR subgroup (*p* = 0.2406). 

The PTR was performed before the chemotherapy in 94.4% of cases, and within this group, 23.0% of the cases were emergency and 77.0% were elective interventions. The rest, 5.6% of PTR, was performed during the chemotherapy, 25.0% as emergency and 75.0% as elective PTR. Comparing the PTR cases before and during the chemotherapy, a significant difference was found only in the localization of the primary tumor (extrapelvic primary tumor 70.3 vs. 18.8%, *p* < 0.0001), while the mPFS1 and mOS values did not differ significantly (mPFS1 301 vs. 340 days; *p* = 0.6875 and mOS 740 vs. 795 days; *p* = 0.7200).

Finally, to emphasize the importance of local care of the mCRC primary tumor, we examined the effect of external beam radiation therapy (EBRT) on the survival of smCRC IPT RSC subgroup patients as well. Within this subgroup, 30.2% of patients received EBRT; the indication for EBRT was urgency (bleeding, obstruction) in 69.2%, and the rest of the radiotherapy cases were mainly elective and prophylactic. Though there were no significant differences between the outcomes of subgroups with or without EBRT, the mOS values showed a clear trend in favor of EBRT (673 vs. 517 days, *p* = 0.0983). Comparing this subgroup to the smCRC PTR RSC subgroup—the same localization, different type of local care—there were no remarkable differences in oncologic outcome (mOS values: 673 vs. 811 days, *p* = 0.5562) (see Figure 3).

## 4. Discussion

The survival of an mCRC patient depends on many factors of different origins. Among these factors, some cannot be, others may be, and some can definitely be influenced by some medical intervention. These factors include tumor-related ones (sidedness, tumor differentiation level, involved metastatic locations, the number of tumor involved sites, the tumor-load of metastatic sites; patient-related data (like sex, age, PS, and even social situation, weight loss, compliance skills); some special laboratory deviations (such as white blood cell count, serum alkaline phosphatase level, serum lactate dehydrogenase level, serum carcinoembryonic antigen level, serum albumin level) and also treatment-related factors (chemotherapy type, the addition of molecularly targeted agents, primary tumor resection, metastasis ablation or resection) [1,6,7].

The distant organ metastasis of smCRC patients confirms that the systemic malignant involvement requires systemic therapy. Despite the need for systemic therapy, the local care of the primary tumor treating or preventing the focal complications cannot be avoided. Achieving a macroscopic tumor-free state by eliminating metastases by any local ablative method also requires PTR. As long as the indication for emergency surgery and the PTR in potentially curable cases is undoubted, PTR for no or mildly symptomatic cases is still under debate despite a significant amount of clinical data and due to the lack of substantial prospective randomized trial results. 

Indications for PTR:

1.Emergency surgery 

The most common indication for emergency surgery in mCRC cases is significant bowel obstruction, with an incidence range of 15% to 29%, meaning 77% of all urgent operations. Gastrointestinal perforation at the site of the primary tumor due to local necrosis or proximal to the primary tumor due to distension is the second most common reason for urgent surgery, with an incidence of 2.6% to 12%. Bleeding, though it is reported in up to 50% of CRC patients, is, in most cases, low volume and rather chronic, and it is a rare cause of emergency surgery [8]. The emergency surgery of mCRC, in contrast to the elective operations, coincides with a lower rate of resection (rectal cancer resection 3.3% vs. 24.3%, *p* < 0.05) and a higher rate of colo/ileostomy formation (21.1% vs. 1.7%, *p* < 0.05) with more extended median hospital stay (18 days vs. 10 days, *p* < 0.05) and a higher rate of postoperative complications (91.1% vs. 23.9%, *p* < 0.05) [9]. The primary outcome as 2-year OS was significantly better in the elective group (80% vs. 42.9%, *p* = 0.002). Still, no significant differences were observed in the 2-year relapse-free survival (RFS) [10]. In certain cases, when the patient is unfit for major surgery, other methods are available to solve the urgent situation, like intestinal stoma formation, or intestinal stent implantation for preventing intestinal obstruction, or intraluminal hemostasis, as well as palliative EBRT for care and prevention of rectal bleeding and focal progression [11,12].

According to our data, 23.2% of all PTRs were emergency surgeries. Comparing the results of emergency and elective surgery within the PTR subgroup, mPFS1 did not show a significant difference (*p* = 0.7794), but mOS were significantly better in the elective PTR subgroup (819 vs. 534 days, *p* = 0.0092; 2 y OS 58.0 vs. 33.9%; OR 0.37 (95% CI 0.19–0.71)), similarly to literary data.

2.Elective PTR for potentially curable patients 

When metastases of mCRC patients are potentially resectable, possible curative treatment can be obtained by surgical resection of the metastases. Patients with oligometastases restricted to the liver or lungs may be candidates for surgical resection. Complete surgical resection of the primary tumor and the metastatic lesions substantially improves overall survival rates to around 35–60% in selected patients. Patients with a better performance status (PS) and better prognosis at baseline (fewer metastatic sites involved, not corrupted organic functions) are more likely to undergo surgery. Complete resection is essential since patients who undergo incomplete resections appear to have similar outcomes to those not resected. Different trials report different rates of metastasectomy, which was performed as only liver metastasectomy, or only lung metastasectomy or both. The complex treatment of mCRC patients with surgery improves DFS and OS considerably [13,14].

The proportion of patients undergoing hepatic metastasectomy has continuously increased from 5% to 19.4% within the last 10 years. The mOS of patients resected for hepatic metastasis is 74.3 months (95% CI 58.5–90.0 months) and their 5-year OS reaches 31–58%, compared to the nonresected patients’ 32.6 months (*p* < 0.0001; HR 0.33 (95% CI 0.22–0.41)). The most appropriate surgical approach is yet to be established. Previously, it was thought that the lesions’ greater number and size meant a worse prognosis; however, if the resection could render R0, the survival is the same. Even repeated hepatectomies in selected patients with recurrent liver metastasis could yield an overall 5-year survival of 30–42%. The postoperative 90-day mortality (4%) and the complication rate (43%, of which 17% are major complications) are still high. Most frequently encountered complications include major bleeding, bile duct injury, perforation of adjacent structures, intra-abdominal infection, wound infection, liver abscess, etc. [15,16].

The role of neoadjuvant and adjuvant chemotherapy is still also debated. Targeted biological agents and locoregional therapies (radiofrequency ablation (RFA), transarterial chemo- or radioembolization (TACE/TARE), and stereotactic radiotherapy (SBRT)) further improve the already favorable results. Neoadjuvant therapy potentially shrinks the tumor and reduces the extent of resection, treats micrometastases, thereby lowering the recurrence rate, and guides further therapeutic plans based on disease response to treatment [15,16,17,18,19]. 

The indications for local therapy of metastatic pulmonary nodules are expanding, with substantial evidence for efficacy in terms of optimal local control for prolonging life and delaying recurrence. Pulmonary metastasectomy might mean anatomical resection, such as lobectomy or segmentectomy and wedge resection, with or without mediastinal lymph node dissection or sampling. Lobectomy has more curative significance for CRC patients with a single pulmonary metastatic lesion ≥ 1.5 cm. Systematic mediastinal lymph node dissection did not improve clinical outcomes for CRC patients who had pulmonary metastasis [20]. OS rates after pulmonary resection were 27–68% at five years, compared to 26.8% of only chemotherapy patients’ survival. In smCRC patients, three parameters were associated with poor survival: multiple lung metastases (HR 2.04 (95% CI 1.72–2.41)); positive hilar and/or mediastinal lymph nodes (HR 1.65 (95% CI 1.35–2.02)); and elevated prethoracotomy carcinoembryonic antigen level (HR 1.91 (95% CI 1.57–2.32)) [21,22,23,24]. 

In selected cases, combining hepatic and pulmonary metastasectomy may also benefit survival with 65.2% five-year OS (95% CI 56.8–72.5) [25]. 

PTR is inevitable for potentially curable patients who are candidates for MA. According to our data, within the PTR subgroup, the MA patients compared to those without MA, mPFS was close to significant (421 vs. 284 days, *p* = 0.0641), but mOS showed an undisputable difference (1122 vs. 616 days, *p* = 0.0003, 2 y OS 82.4 vs. 43.8%, OR 0.17 (95% CI 0.08–0.36). The difference between PTR with MA and IPT subgroups is even more significant (mPFS1 421 vs. 259 days, *p* = 0.0004, 1 y PFS OR 0.27 (95% CI 0.14–0.50) and mOS 1122 vs. 495 days, *p* < 0.0001, 2 y OS 82.4% vs. 26.9%, OR 0.08 (95% CI 0.03–0.18).

3.PTR for patients with no or mild symptoms from the primary tumor
IArguments against PTR

*(1) PTR has high postoperative morbidity and mortality.* One of the strongest counterarguments against PTR is postoperative morbidity and mortality. Perioperative complications after PTR appear within the first 30 days in 2.0% to 48.3% of cases and are significantly more frequently observed after rectal surgery than colon surgery (40.3% vs. 19.9%, *p* = 0.001). Considering both colon and rectal interventions, the postoperative surgical and medical complication rates were 35.6% and 25.3%, respectively. These complications are anastomotic leakage (2.9 to 15.3%), ileus, surgical site infection (2–25%), hernia/dehiscence (0.5–2.6%), stoma complications (ileostomy 35.6%, colostomy 21.8%), postoperative hemorrhage, deep vein thrombosis, pulmonary embolism, pulmonary infection, renal failure, cerebral insult, myocardial infarction, etc. [13,26,27,28,29]. 

Postoperative mortality ranges between 0 and 11.7% (median rate of 4.6–6.5%), and a bit higher for colon cancer than rectal cancer (5.1% vs. 3.9%); the frequencies were in emergency cases 27.8%, and in elective surgery, 7.3% (*p* = 0.002). Nevertheless, postoperative mortality is frequently the direct result of pre-existing comorbidity and not always the direct result of the surgical procedure (56% vs. 44%, *p* = 0.001). At least one-third of mortality after colorectal surgery is attributed to leaks. The 30-day mortality rates after PTR are higher than reported for elective surgery in stage I–III patients [13,30,31,32].

The unmodifiable risk factors of postoperative morbidity and mortality of PTR are age and frailty (although morbidity and mortality rates in elderly patients could be similar to their younger counterparts who are undergoing elective surgery; however, these rates could be up to nine times higher in cases of emergency surgery); male patients; rectal cancer; hepatic tumor load > 50%; and prior abdominal surgery, probably due to adhesions. The other, modifiable risk factors are the presence of comorbid illnesses (mainly cardiovascular and pulmonary disease, diabetes, and obesity); a patient’s general physical condition; surgeons’ experience; hospital case load and surgical facilities [28,33]. 

*(2) The delay of chemotherapy results in worse oncologic outcomes.* In adjuvant systemic treatment of CRC, even a four-week delay of cancer treatment is associated with increased mortality (HR 1.09 (95% CI 1.01–1.12)) [34]. Similar observations were presented in mCRC cases by a large number of retrospective cohort studies, where, according to diagnosis-to-treatment-interval (DTI) subgroups (<30 days, 31–150 days, and ≥151 days), for the longer-time intervention groups compared to the <30 days subgroup, the risk of death was increased (1.37 (95% CI 1.28–1.47), *p* < 0.0001 and 1.36 (95% CI 1.25– 1.47), *p* = 0.0001, respectively), with no difference between the two longer DTI subgroups [35]. Moreover, prolonged delays in treatment could cause cancer anxiety and worry in patients and their families. Anxiety may be heightened by the commonly held view that delays of even a few weeks can adversely affect outcomes [36]. Conclusively, the chemotherapy-free perioperative time interval could have a negative impact on the effectiveness of systemic therapies. 

Our study proved the opposite. The oncologic outcomes, mPFS1, and mOS were significantly better in the PTR subgroup compared to the IPT subgroup (mPFS1 301 vs. 259 days, *p* < 0.0001, and mOS 760 vs. 495 days, *p* < 0.0001). To assess the real value of PTR, we eliminated all factors showing significant differences between the subgroups. From both PTR and IPT subgroups, all patients were excluded who received MTA and had MA and whose PS was worse than ECOG 1. The difference was still strongly significant in favor of PTR, mPFS1 287 vs. 221 days, *p* = 0.0008, 1 y PFS 35.6 vs. 21.1%, OR 0.48 (95% CI 0.24–0.99), and mOS 675 vs. 459 days, *p* = 0.0009, 2 y OS 45.9 vs. 24.1%, OR 0.37 (95% CI 0.18–0.79).

*(3) PTR does not improve OS.* Based on retrospective data analysis and meta-analysis, some authors declared that the survival of smCRC patients was not significantly different between PTR and IPT groups (*p* = 0.95). Their opinion is that PTR in an unselected mCRC population does not improve survival and is associated with a high risk of postoperative mortality [37]. In addition, resection does not significantly reduce the risk of complications from the primary tumor (i.e., obstruction, perforation, or bleeding) [38]. Meanwhile, statistically insignificant or clinically irrelevant differences in QoL and fatigue were observed when PTR was added to systemic therapy. Patients in the PTR arm experienced serious adverse events (SAEs) twice as often as patients in the standard arm (*p* < 0.001) [39]. Another meta-analysis reported that among mCRC patients with an asymptomatic primary tumor, leaving the primary tumor intact did not cause unacceptable complications, and survival was not significantly compromised [40]. 

In our study, PTR was able to improve mOS (see Discussion 3.I.(2)). PTR not only results in significant mOS advantage, as seen in the comparison of reduced subgroups (*p* = 0.0009), but also creates the necessary circumstances for MA to further improve OS (see Discussion 2).

*(4) Chemotherapy can control the primary tumor without elective PTR.* Continuing the correlations above and according to some authors’ observations, there is no statistically significant OS difference between smCRC patients with elective vs. urgent PTR (*p* = 0.052). However, there is a trend towards better 2-year OS in the elective group (74.3 vs. 37.5%) [10]. Another opinion is that most IPT smCRC patients (89%) never required palliation of their primary tumor, owing to the general chemotherapy effect. Only 6% required emergent surgery for primary tumor obstruction or perforation, and 9% required nonoperative intervention (i.e., stent or radiotherapy). Conclusively, the low incidence of late, symptom-directed intervention does not justify the routine use of prophylactic surgery or radiotherapy. Therefore, surgery-related morbidity and mortality should be avoided. Additionally, investigators demonstrate that most patients with smCRC who receive upfront systemic therapy never require palliative surgery [32]. In 70% of patients who received systemic treatment before PTR, major histological tumor regression was observed, suggesting that initial chemotherapy can control the primary tumor in most patients [41].

*(5) PTR can elevate the vascular density of metastases.* Circumstantial evidence shows an increased growth rate of liver metastases after PTR, which is determined by an increased vascular density, proliferation rate, and metabolic growth rate initiated by the surgical intervention. These data suggest that the outgrowth of metastatic disease may at least partially be controlled by the primary tumor [42]. Both peritumoral and intratumoral vascular density were elevated in synchronous metastases from patients with PTR compared to patients with IPT. PTR results in an increased vascularization of metastatic lesions. Liver metastases have highly to very highly vascularized peritumoral area in PTR patients (63%, compared to 29% in IPT patients (*p* = 0.025)) [43]. 


IIArguments for PTR


*(1) Longer OS with PTR.* In light of the lack of robust clinical evidence, the treatment decisions differ from one country to another and from one institute to another. As for the United States, most patients with smCRC undergo PTR; in contrast, in The Netherlands, a trend toward a nonresection approach has been observed [42]. In different centers, the rate of smCRC patients treated firstly with PTR ranges from 29 (– 57.4 – 66 –) to 72% [37,44,45,46]. Nonetheless, many clinical data, retrospective analyses, and meta-analyses prove the significant effect of PTR on smCRC patients’ survival. In these publications, the available mOS values with PTR range from a maximum of 30.7 months (vs. 21.9 months (*p* = 0.031) [13] to the minimum of 16 (vs. 9 months (*p* < 0.001) [47]. All publications showed a highly significant difference between the PTR and IPT groups. After PTR, patients lived longer, 6–9 months, compared to those who did not have PTR [13,48]. After excluding the patients who underwent metastasectomy, the mOS of patients who underwent PTR was 15.2 months, compared with 8.3 months if they did not have surgery (*p* < 0.0001) [49]. Other authors analyzed the effect of PTR on survival based on different aspects. There was no significant difference in OS between emergency and elective surgery subgroups (22.9 vs. 16.1 months, *p* = 0.9) [50]. After PTR, no difference was registered in the response rates to chemotherapy (HR 0.85 (95% CI 0.40–1.80); *p* = 0.662) [48]. However, patients who underwent PTR were less likely to have rectal cancer (OR 0.49 (95% CI 0.39–0.63); *p* < 0.001) [51]. In other studies, Cox proportional analysis revealed that the use of chemotherapy (HR 0.47 (95% CI 0.41–0.54)), PTR (HR 0.49 (95% CI 0.41–0.58)), second-line chemotherapy (HR 0.47 (95% CI 0.45–0.64)), and metastasectomy (HR 0.54 (95% CI 0.45–0.64)) were correlated with superior survival [49]. Univariate analysis identified three significant prognostic variables (number of distant sites involved, metastases to liver only, and volume of hepatic replacement by tumor) in the resected group [47]. In multivariate analysis, PTR was the most substantial independent prognostic factor of OS (HR 0.4 (95% CI 0.3–0.6); *p* < 0.0001) [45]. By the recommendations of some authors, upfront resection should be performed in patients with “favorable” oncologic criteria (e.g., metachronous lesions, fewer metastases, unilobar disease, no extrahepatic disease) [52]. 

In our study, the rate for elective primary PTR was 48.8% (literary data: 29–72%). In the PTR subgroup, the mOS was 25.0 months (literary data: 16–30.7 months) compared to the 16.3 months in the IPT subgroup, with a survival benefit of 8.7 months (literary data: 6–9 months). In the reduced subgroups (no MTA, no MA, and no PS ECOG > 1), these data were 22.2 vs. 15.1 months, with a 7.1-month advantage. 

*(2) The delay of chemotherapy due to PTR does not affect OS.* Based on more clinical studies focusing on the timing of the primary systemic treatment of mCRC, many authors did not find a correlation between the delay of immediate systemic treatment and OS [53]. In a real-world cohort of patients treated within eight weeks of diagnosis, time without treatment (TWT) did not have a negative impact on survival outcomes in mCRC. In the 4–8-week, 2–4-week, and under-two-week subgroups, mOS were 26.9, 22.6, and 18.05 months (*p* < 0.0001). Moreover, the longer TWT was associated with a lower hazard of death (HR 0.78 (95% CI 0.73–0.84), *p* < 0.0001) compared to the 2–4-week subgroup. The under-two-week subgroup showed the highest hazard of death (HR 1.26 (95% CI 1.14–1.38) *p* < 0.0001) [54]. TWT in cases of PTR was a median of 35–95 days and was not associated with shorter survival (HR 1.17 (95% CI 0.93–1.46), *p* = 0.4898) [51,55]. The summary of several studies suggests that both PTR and primary chemotherapy survival outcomes are equivalent, with a trend towards an OS advantage with PTR. In this cohort, PTR was relatively safe, with transient morbidities, and the vast majority of patients were able to proceed with systemic chemotherapy [56]. While long delays are obviously undesirable, the tight waiting times may curtail preparation for major surgery, meaning underusing resources that may affect the outcome. Conclusively, longer delays are not associated with poorer survival [36]. 

PTR was carried out as elective surgical intervention in 76.8% of all PTR cases, causing measurable delay in the beginning of systemic therapy. According to the data presented (see Discussion 3.I.(2)), this delay did not negatively affect OS. 

*(3) PTR reduces the tumor burden (TB) of the patient.* The TB of mCRC patients consists of the primary tumor and the tumorous involvement of metastatic organs. The reduction in tumor volume prevents tumor-related symptoms, increases chemotherapy effectiveness [10], and affects the malignant metabolic and immunological alterations. The lower TB correlates with better PS (*p* = 0.0002) and superior mOS (low TB vs. high TB 22.7 vs. 11.9 months, *p* = 0.0017) [57]. Chemotherapy may reduce TB, particularly if major organ function is not compromised. Patients with compromised PS resulting from a high TB are more likely to experience a survival benefit from chemotherapy (*p* < 0.001, OR 1.82 (95% CI 1.63–2.02)) [58]. 

The connection between high neutrophil-to-lymphocyte ratio (NLR) as a biomarker of systemic inflammation and poor outcomes in cancer patients is well known. As an inflammatory response, neutrophilia inhibits the immune system by suppressing the cytolytic activity of immune cells such as lymphocytes, activated T cells, and natural killer cells [59]. There is a strong correlation between primary tumor volume and NLR (*p* < 0.001); an elevated pretherapeutic NLR was associated with inferior DFS and poor pathological response to neoadjuvant radiochemotherapy [60]. In mCRC cases, high pre- and low post-PTR NLR reversal was associated with significantly improved OS (HR 0.53; *p* = 0.017) [61].

*(4) Increased need for PTR over time.* As the incidence of CRC increases continuously (0.5–1.0% per year), so too does the incidence of upfront metastatic cases (a 5% increase in 20 years) as well. Approximately 22–29% of patients present with metastases at the first medical examination [3,62]. The introduction of sophisticated multimodal treatments elevated the mOS from 18 to over 30 months. The prolonged survival also means more chance of facing complications from the primary tumor. On the other hand, surgical metastasectomy and other metastasis ablative methods brought new perspectives on survival. For example, liver surgery increased from 4% to 10% in 10 years [63]. The effect on survival of any intervention on metastases is based on a preceding or simultaneous PTR. Thus, in cases of potentially resectable metastases of smCRC, the demand for PTR has increased over time. 

According to our data, the rate of MA in the two subgroups differed significantly: 21.8% vs. 1.2%, *p* < 0.0001. However, we have to admit that the indication for PTR among those patients who are potential candidates for MA is more frequent, as an inevitable step for macroscopic tumor-free condition. Within the PTR subgroup, the MA patients had significantly better mOS (*p* = 0.0003) (see Discussion 2).

*(5) Higher risk of perioperative morbidity and mortality of emergency surgical interventions during chemotherapy.* During chemotherapy, the most frequent therapy-connected side effect is neutropenia, the frequency of which (NCI grade 3–4 hematologic toxicity) in doublet therapies ranges between 41.7 and 53.8% [64,65]. Even nowadays, its life-threatening form, febrile neutropenia (FN), appears approximately in 13.7% [66] and predicts early and overall mortality (HR 1.15 and 1.35, respectively) (9.0 deaths per 1000 person-months of treatment) [67]. Rates of 30-day mortality significantly vary among the mild, moderate, and severe neutropenic subgroups (*p* = 0.02) with 2.9%, 16.2%, and 27.6%, respectively [68]. Ongoing neutropenia raises perioperative morbidity and mortality rates of urgent surgical interventions. Major postoperative complications occurred significantly more frequently in chemotherapy patients (44–45% vs. 27–39%, *p* = 0.033), and their mortality was also significantly higher than the mortality of nonchemotherapy patients (22% vs. 10%, *p* < 0.001) [69]. 

The use of bevacizumab, a vascular endothelial growth factor A inhibitor monoclonal antibody (VEGF-Ai) combined with chemotherapy doublets, is one of the backbones of mCRC treatment [70]. Bevacizumab dose-dependently and significantly increases the overall risk of hemorrhage by 5.8% (HR 1.96 (95% CI 1.27–3.02) [71]. A total of 28.6% of the patients receiving bevacizumab and thromboprophylaxis (apixaban, aspirin, etc.) simultaneously experience an adverse bleeding event that results in some form of treatment discontinuation [72]. Patients treated with bevacizumab have an increased risk of postoperative bleeding (especially tumor-associated hemorrhage) in 10–20% of cases [73]. Compared with routine therapy, bevacizumab increased the incidence of wound-healing complications; the pooled estimate of OR is 2.32 (95% CI 1.43–3.75) (*p* < 0.001) [74].

*(6) PTR prevents primary tumor-associated complications.* The IPT is a possible place of complications requiring intervention in patients undergoing chemotherapy; its frequency varies from 3.5% to 40%. The mean time to the onset of these complications ranges from 3 to 11 months. These primary tumor-related local complications are lower gastrointestinal bleeding, bowel obstruction with ileus/subileus, tumor perforation, diarrhea, and incontinence. Nevertheless, intact primary tumors may cause systemic complications as well, including weight loss, anorexia, nutritional depletion, and pain. These complications often warrant emergency surgery, which has a higher rate of perioperative mortality and morbidity than elective surgery. This may be more challenging when the patient has myelosuppression due to systemic chemotherapy. The complex challenge is not only the high-risk emergency intervention but also the lasting inconveniences that cause significant deterioration of QoL [29,56]. Comparative clinical studies proved the otherwise implicit difference between PTR and IPT subgroups of smCRC patients. The incidence of grade 3–4 nausea, vomiting, and ileus occurred significantly more frequently in the IPT subgroup with 9 vs. 3% for nausea (*p* = 0.004), 9 vs. 4% for vomiting (*p* = 0.043) and 8 vs. 3% for ileus (*p* = 0.019), respectively [42].

*(7) Chemotherapy is unable to control the primary tumor.* With sophisticated systemic therapies, an over-two-year survival is frequently available in mCRC. Meanwhile, in IPT smCRC patients, primary tumor-related symptoms, most commonly obstruction, ranged from 3% to 46% [56]. Many studies proved that most of the mCRC patients had no complete histological response to chemotherapy, considering both the primary tumor and the metastases. Major histological tumor regression (TRG2) was observed in 70% of patients [41].

*(8) Chemotherapy can facilitate further dissemination and the development of drug resistance.* Robust preclinical evidence indicates that chemotherapy can induce intratumoral and systemic changes that paradoxically promote cancer cell survival/proliferation and dissemination under certain circumstances. Via different extracellular and intracellular mechanisms, chemotherapy supports the development of chemoresistance, neoangiogenesis, and tumor cell intravasation, which may sabotage the final effect of cancer care [75,76]. These findings anticipate the definitely limited effect of chemotherapy courses and also affirm the role of local interventions.

*(9) PTR ensures more precise staging.* An argument favoring PTR is the more accurate staging of the peritoneal cavity. The presence of peritoneal metastases in mCRC patients is associated with poorer prognosis. The individual peritoneal tumor nodules are frequently below the detection level of conventional CT or PET due to their small size and limited contrast resolution in soft tissues. The true incidence of peritoneal metastases is unclear, although in autopsy series, it was reported to be as high as 40–80%. The benefit of systemic chemotherapy is dramatically reduced in the subgroup of CRC patients with peritoneal metastases. Exploratory laparoscopy/laparotomy can survey the peritoneal cavity to separate the potentially curable patients from those who need special medical attention, possibly requiring hyperthermic intraperitoneal chemotherapy (HIPEC) [42,77]. 

## 5. Conclusions

The role of PTR in patients with smCRC with no curative intent is still under debate. Our recent observations on the role of PTR in smCRC patients over the implicit results justified by the available literature data showed some extraordinary, unexpected findings that can influence our future strategies. Regarding the unambiguous role of PTR in the smCRC cohort, the outcome results of the PTR subgroup were very similar to the mmCRC group’s, emphasizing the fact that the absence of primary tumor characterized both patient groups. To our knowledge, this is the first description of this phenomenon, which has both clinical and tumor biological significance. Surprisingly, the comparison between the two types of local care of smCRC RSC, surgery vs. irradiation, did not show a significant difference in OS. It seems that for any manipulating of the primary tumor of smCRC patients, both forms of local care may be beneficial for improving survival.

Finally, some limitations of this study should be emphasized. First, as a retrospective observational study based on the clinical routine in a quality-controlled, solitary oncology center, we cannot exclude the individual treatment decisions based on the treating physician’s experience, attitude, and foresight. Another important limitation of this paper is the lack of molecular information in this group of patients, which has an important role in today’s decision making in oncology care. 

Conclusively, our study shows several new aspects of the role of PTR in the treatment process of smCRC patients. These results attend to the novel current literary data [78], though further investigations, first of all, prospective randomized multicenter clinical trials, are needed to clarify this question. 

## Figures and Tables

**Figure 1 cancers-16-01460-f001:**
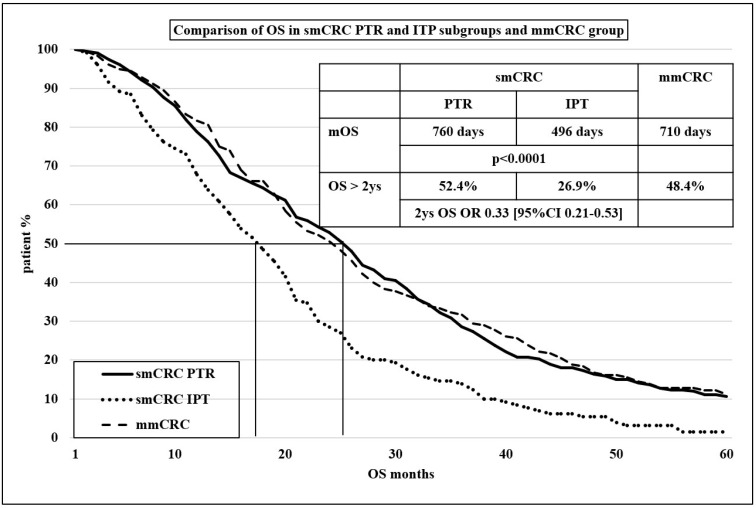
Kaplan–Meier curve for comparison of OS in smCRC PTR and IPT subgroups and mmCRC group.

**Figure 2 cancers-16-01460-f002:**
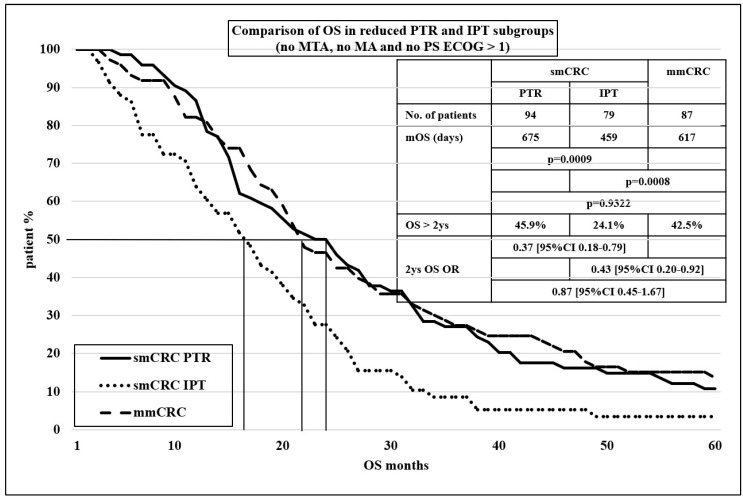
Kaplan–Meier curve for comparison of OS in reduced PTR and IPT subgroups (no MTA, no MA was used and no patient PS ECOG > 1).

**Figure 3 cancers-16-01460-f003:**
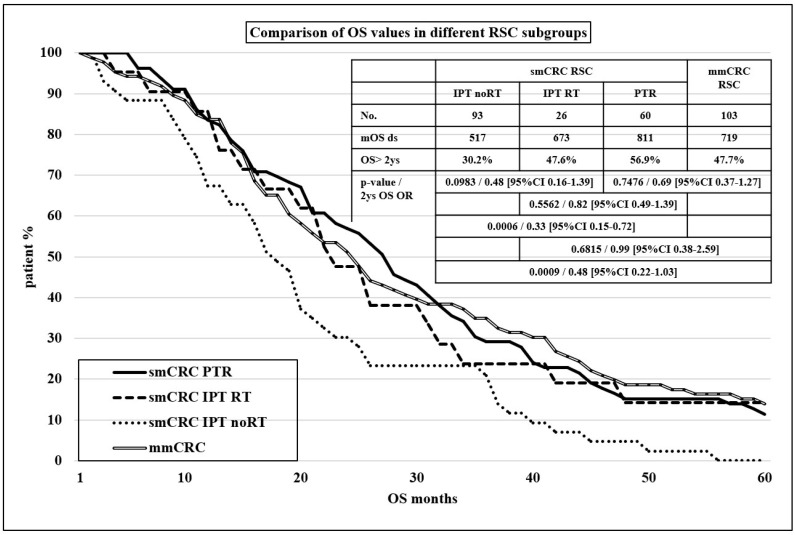
Kaplan–Meier curves for comparison of OS values in different RSC subgroups.

**Table 1 cancers-16-01460-t001:** Patients’ characteristics.

	smCRC		mmCRC
	PTR	IPT	
No. of patients	285	164	215
Male/female rate	166/119 (58.2%/41.8%)	94/70 (57.3%/42.7%)	128/87 (59.5%/40.5%)
Age (median)	65.7 (30.1–85.6)	64.0 (31.0–92.7)	67.1 (26.2–85.4)
PS (average) (ECOG)	0.98	1.1	1.04
	*p* = 0.0456		
RCC	74 (25.9%)	28 (17.1%)	40 (18.6%)
	*p* = 0.0243		
LCC	118 (41.4%)	50 (30.5%)	72 (33.5%)
	*p* = 0.0213		
RSC	93 (32.6%)	86 (52.4%)	103 (47.9%)
	*p* < 0.0001		
Extrapelvic vs. intrapelvic primary tumor	67.4%	47.6%	
	*p* < 0.0001		
Mono-/multiorgan metastases	216/69 (75.8%/24.2%)	103/61 (62.8%/37.2%)	168/47 (78.1%/21.9%)
	*p* = 0.0034		

**Table 2 cancers-16-01460-t002:** A comparison of first-line systemic therapies and metastasis ablation in the different groups.

	smCRC		mmCRC
	PTR	IPT	
Chemotherapy			
Monotherapy (FP)	11.6%	13.4%	13.1%
	*p* = 0.5688		
Doublet therapy	88.4%	86.6%	86.9%
IRI/OXA	96.0%/4.0%	95.8%/4.2%	93.6%/6.4%
5FU CI/CAP	96.4%/3.6%	96.5%/3.5%	87.8%/12.2%
Doublet therapy + MTA	45.3%	28.7%	31.6%
	*p* = 0.0005		
VEGFi/EGFRi	70.5%/29.5%	59.6%/40.4%	61.8%/38.2%
Metastasis ablation	62 (21.8%)	2 (1.2%)	58 (26.9%)
	*p* < 0.0001		

**Table 3 cancers-16-01460-t003:** The survival results of first-line systemic therapy.

	smCRC		mmCRC
	PTR	IPT	
Progression-free survival 1	301	259	273
	*p* < 0.0001		
1 year PFS	39.2%	26.6%	36.8%
	OR 0.56 (95% CI 0.36–0.87]		
Overall survival	760	495	710
	*p* < 0.0001		
2 y OS	52.4%	26.9%	48.4%
	OR 0.33 (95% CI 0.21–0.53]		

**Table 4 cancers-16-01460-t004:** PTR and IPT subgroup survival depending on the number of metastatically involved organs.

	PTR		IPT	
Metastases	Monoorganic	Multiorganic	Monoorganic	Multiorganic
No. of patients	68	26	50	29
OS (days)	732	422	463	371
	*p* = 0.0045		*p* = 0.5089	
	*p* = 0.3344			
		*p* = 0.0017		
2 y OS	50.9%	33.3%	23.1%	26.3%
	OR 0.48 (95% CI 0.17–1.38)		OR 0.84 (95% CI 0.24–2.97)	
	OR 0.29 (95% CI 0.12–0.72)			
		OR 0.71 (95% CI 0.18–2.80)		

**Table 5 cancers-16-01460-t005:** (**A**) PTR and IPT subgroup survival depending on primary tumor localization. (**B**) smCRC patients’ reduced subgroup survival depending on primary tumor localization.

**(A)**								
	**PTR**				**IPT**			
Primary tumor	RCC	LCC		RSC	RCC	LCC		RSC
No. of patients	74	118		93	28	50		86
	26.0%	41.4%		32.5%	17.1%	30.5%		52.4%
mOS (days)	552	818		811	376	436		584
	*p* = 0.3487				*p* = 0.8611			
		0.7134				*p* = 0.0049		
		*p* = 0.2299				*p* = 0.0419		
2 y OS	36.8%	58.2%		56.9%	22.7%	15.9%		35.9%
	OR 0.42 (95% CI 0.21–0.83)				OR 0.64 (95% CI 0.18–2.32)			
		OR 0.95 (95% CI 0.52–1.75)				OR 0.34 (95% CI 0.13–0.88)		
	OR 0.44 (95% CI 0.21–0.89)				OR 0.52 (95% CI 0.17–1.61)			
**(B)**								
**Primary Tumor**	**E** **xtrapelvic**		**In** **trapelvic**		**Extrapelvic**		**Intrapelvic**	
No. of patients	70		24		40		39	
mOS (days)	680		656		361		644	
	*p* = 0.7496				*p* = 0.0070			
	*p* < 0.0001							
			*p* = 0.3004					
2 y OS	46.2%		45.5%		9.1%		44.0%	
	OR 0.97 (95% CI 0.36–2.64)				OR 0.13 (95% CI 0.03–0.53)			
	OR 0.12 (95% CI 0.03–0.43)							
			OR 0.94 (95% CI 0.30–2.98)					

**Table 6 cancers-16-01460-t006:** Major differences between PTR and IPT subgroups.

	smCRC		mmCRC
	PTR	IPT	
PS (average) (ECOG)	0.98	1.1	1.04
	*p* = 0.0456		
RCC	25.9%	17.1%	18.6%
	*p* = 0.0243		
LCC	41.4%	30.5%	33.5%
	*p* = 0.0192		
RSC	32.6%	52.4%	47.9%
	*p* < 0.0001		
Extrapelvic primary tumor	67.4%	47.6%	52.1%
	*p* < 0.0001		
Monoorganic metastasis	75.8%	62.8%	52.1%
	*p* = 0.0034		
Doublet therapy + MTA	45.3%	28.7%	31.6%
	*p* = 0.0005		
Metastasis ablation	21.8%	1.2%	26.9%
	*p* < 0.0001		

## Data Availability

The data presented in this study are available on request from the corresponding author. The data are not publicly available due to personal data protection.

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
