# Peer review of "The Real-Life Impact of Primary Tumor Resection of Synchronous Metastatic Colorectal Cancer—From a Clinical Oncologic Point of View"

_cancers, 2024, doi:10.3390/cancers16081460_

Round 1

Reviewer 1 Report

Comments and Suggestions for Authors

General points:

This research was conducted based on the clinical question focusing on the role of PTR (primary tumor resection) in the treatment of synchronous metastatic colorectal cancer (smCRC).  

I believe that the authors have made significant efforts to elaborate their extensive data analyses: however, since this work has been submitted as an original article, the manuscript musta be more structured, summarized, and sophisticated to effectively show the authors’ findings and viewpoints, as well as to draw readers’ attention to this research topic.

I would like to recommend the authors to carefully read the journal guidelines and “how to write” articles before moving to the next review process.

Specific points:

1.      The title must be more scientifically renamed: “at least something” must be more specific.

2.      Introduction: Any epidemiological data must be accompanied by appropriate citations.

3.      Method: The methodology did not reflect the extensive data shown in the following results section. In the results section, many subgroup analyses have been conducted.

4.      Method: There were some obscure definitions. Why did the authors compare “1y PFS” or “2ys OS”? Why did they use “mmCRC” as the control group? (This was not clearly defined, either)

5.      Results: It is very hard to interpret the results because many complex subgroup analyses (that were not defined in the methods). I would recommend the authors to divide the results into subsections.

6.      Discussion: This section is felt like “narrative review of the literature”. The discussion section must include the authors’ implications of the study based on the observations.

7.      Conclusions: This section can be more shortened and reshaped including take-home message and future directions. 

Comments on the Quality of English Language

More sophisticated achademic writing may be needed.

Author Response

Answers to Reviewer 1.

Dear Collegue,

Thank You very much for Your observations and advices. We appreciate every help to make our paper better, more understandable becose we think that the subject of the paper is very important and needed in the everyday practise.

Answers:

  1. The title must be more scientifically renamed: “at least something” must be more specific.

„At least something” is not a scientifical part of the title, rather a „literary” which shows our opinion - that is supported by the presented data - that „something” – either PTR or EBRT – must be done to improve survival. 

  1. Introduction: Any epidemiological data must be accompanied by appropriate citations.

All data in the Introduction comes from the References presented in or after each paragraph.

  1. Method: The methodology did not reflect the extensive data shown in the following results section. In the results section, many subgroup analyses have been conducted.

I understand that the proper explanation for these complex comparisons between many subgroups is not well transparent, so I tried to replace it. Hopefully, this explanation together with the subsections You suggested will make it properly understandable.

  1. Method: There were some obscure definitions. Why did the authors compare “1y PFS” or “2ys OS”? Why did they use “mmCRC” as the control group? (This was not clearly defined, either)

The 1y PFS an 2 ys OS was used, because most chemotherapy combination used as 1st line treatment for mCRC has approximately 1y mPFS, and mCRC patients has approximately 2ys mOS. We thought that these two data can represent well the effect of PTR on PFS and OS. Originally we used more data to compare but it made this quite complex paper even more complex and less transparent. I put the explanation of use of mmCRC subgroup as the controll group. We think that the two subgroups smCRC with PTR and mmCRC (already without primary tumor) may behave similarly as both group is without the primary tumor. And the results showed strong similarity between these two subgroups.

  1. Results: It is very hard to interpret the results because many complex subgroup analyses (that were not defined in the methods). I would recommend the authors to divide the results into subsections.

Thank You very much for these observation. Your objectivity helps us to make our paper more transparent. I made more modification in Method and Result section, hopeing that it made it more clear and understandable.

  1. Discussion: This section is felt like “narrative review of the literature”. The discussion section must include the authors’ implications of the study based on the observations.

As you said, the Discussion section is mainly a „narrative review of the literature”. We aimed to collect as many pros and cons as possible so that PTR could see the problem from many points of view. Some opinions are somewhat experimental or theoretical, that may not be directly connected to our clinical data. This is why we did not want to „mix” Results with Discussion. All our data with precise outcome results are available in the Results section, showing that we are for PTR or at least an EBRT for rectal cancer.

  1. Conclusions: This section can be more shortened and reshaped including take-home message and future directions. I tried to make the Conslusions more focused, shorter with a take-home massege.

Best wishes,

Balázs Pécsi

Reviewer 2 Report

Comments and Suggestions for Authors

Interesting paper concerning the role of Primary Tumor resection in Synchronous metastastic CRC.

General considerations
No mentions regarding survival results after  complete and not complete PTR.  This aspect could change survival results.

No mention regarding peritoneal carcinosis and treatment modality (10% of synchronous metsastatic colorectal cancer)

No mentions of the different ablative modality (MA)

Corrections

Tab 1: Patients characteristics

Tab 3: the survival results....

Tab 4: analysis (Survival )

Tab 5 A and B: analysis (Survival)

Discussion: the period between...the distant organ metastasis......substantial prospective randomized trial results, should be removed (not necessary).

Comments on the Quality of English Language

Discussion paragraph: the level of english language is not proper and fine, comprehension of readers is not simple and a native translator should be involved.

Author Response

Answers to Reviewer 2

Dear Collegue,

Thank You very much for Your observations and advices. We appreciate every help to make our paper better, more understandable becose we think that the subject of the paper is very important and needed in the everyday practise.

Answers:

No mentions regarding survival results after  complete and not complete PTR.  This aspect could change survival results.

These data are temporarily unavailable. We accepted PTR as successful even in R2 resections. According to our point of view even with R2 resection the two major complication (bleeding, obstruction) were preventable, and probably a the majority of the primary tumor-mass was removed. Probably the fundamental difference between complete and incomplete PTR is only observable in cases with MA. Without MA only the volume of residual tumor is different. Other explanation: that part of smCRC IPT group who received EBRT on the primary tumor – which migh be considered as R2 resection, with survival tumor cells – had similar OS as PTR group. Despite this explanation, I must admit that you gave me a good idea of how to analyse this group of patients further.

No mention regarding peritoneal carcinosis and treatment modality (10% of synchronous metastatic colorectal cancer)

The rate of peritoneal carcinosis in the PTR and IPT groups was 16.8 vs. 22.6 % (p=0.1370). There was no significant difference, though a strong peritoneal carcinosis is contraindication for a not necessary resection. We considered that  PTR in smCRC is the „only” tumor-reductive and complication-preventing method, so peritoneal carcinosis had no significant effect on the outcome. (it may has an infuence on the healing of the anastomosis). It did not even cause a higher rate of postoperative complications.

No mentions of the different ablative modality (MA)

Almost 95% of all MA was surgical, the rest in equal parts SBRT and RFA. Due to the very low number of non-surgical MA, the comparison of the different MA methods was not possible. From this paper’s point of view, the only emphasised data is the significant difference in MA rate between the PTR and IPT groups. Together with our mmCRC patients’ data we are planning an other paper that focuses on MA question.

Corrections

All suggested correction is done and attached in ppt format as MDPI required.

Discussion: the period between...the distant organ metastasis......substantial prospective randomized trial results, should be removed (not necessary).

It’s just a short „summary” about the need of PTR in smCRC. If You don’t mind, I’d like to keep it.

Comments on the Quality of English Language

Discussion paragraph: the level of english language is not proper and fine, comprehension of readers is not simple and a native translator should be involved.

The paper was finally reviewed by a native (but not medical) lector. I contacted MDPI about improving the English language of the paper. However, I have to mention that some months before, the same team produced another paper that was accepted with the original English text.

Best wishes,

Balázs Pécsi

Round 2

Reviewer 1 Report

Comments and Suggestions for Authors

Thank you very much for responding to the queries. The method and results sections has been more transparent as expected.

1.      Regarding the title, the authors are advised to read the “Instructions for Authors’ instruction” available at https://www.mdpi.com/journal/cancers/instructions. According to the instruction, “the title of your manuscript should be concise, specific and relevant”. That is why “more scientific” title has been recommended at the first-round review”. The authors’ (non-scientific) opinions can be included to the discussion or conclusion section.  

2.      Regarding the contents of discussion, the authors are advised to read the “Instruction for Authors” available at https://www.mdpi.com/journal/cancers/instructions. According to the instruction, “Authors should discuss the results and how they can be interpreted in perspective of previous studies and of the working hypotheses. The findings and their implications should be discussed in the broadest context possible and limitations of the work highlighted. Future research directions may also be mentioned. This section may be combined with Results.” That is why the comment on the first-round review recommended to include the authors’ implications of the study based on the observations. Doing so is not “mixing the results with discussion”.

3.      Introduction: In general, the citations should be appeared at the end of the relevant sentences. In this manuscript, citations 1, 2, and 3 appeared at the end of second paragraph. However, the data from previous reports were shown within line 1 to 4 in that paragraph.  

Comments on the Quality of English Language

The English language is appropriate.

Author Response

Dear College,

Thank You very much for Your help in the improvement our paper. I already read the “Instructions for Authors” many times. I accepted Your opinion about the title, the citations and the contents of discussion. I hope I understood Your advices properly and the result is close to what is needed.

  1. Regarding the title, the authors are advised to read the “Instructions for Authors’ instruction” available at https://www.mdpi.com/journal/cancers/instructions.According to the instruction, “the title of your manuscript should be concise, specific and relevant”. That is why “more scientific” title has been recommended at the first-round review”. The authors’ (non-scientific) opinions can be included to the discussion or conclusion section.  

I also read these Instructions. I thought that after a “concise, specific and relevant” part of the Title, a bit personal and less scientific part of the title is permissible. According to Your suggestion, I deleted this part. 

  1. Regarding the contents of discussion, the authors are advised to read the “Instruction for Authors” available at https://www.mdpi.com/journal/cancers/instructions.According to the instruction, “Authors should discuss the results and how they can be interpreted in perspective of previous studies and of the working hypotheses. The findings and their implications should be discussed in the broadest context possible and limitations of the work highlighted. Future research directions may also be mentioned. This section may be combined with Results.” That is why the comment on the first-round review recommended to include the authors’ implications of the study based on the observations. Doing so is not “mixing the results with discussion”.

I attached our own data and observations to the paragraphs of Discussion where it was connectable. Some paragraphs contained experimental opinions or data that we are not investigated (surgical outcome, surgical complications, etc.), they remained without comment. I really hope that this is the completion what You adviced. 

  1. Introduction: In general, the citations should be appeared at the end of the relevant sentences. In this manuscript, citations 1, 2, and 3 appeared at the end of second paragraph. However, the data from previous reports were shown within line 1 to 4 in that paragraph.  

Yes, I understand. Corrected.

Best wishes,

Balázs Pécsi

Reviewer 2 Report

Comments and Suggestions for Authors

Interesting paper concerning the role of Primary tumour resection in mildly or asymptomatic sinchronous metastastases. 

Corrections were correctly done and conclusions are significative

Round 3

Reviewer 1 Report

Comments and Suggestions for Authors

The structure of the manuscript has been improved and the authors have very well responded to the queries.